# Dynamic metabolome profiling uncovers potential TOR signaling genes

Stella Reichling[1], Peter F Doubleday[1], Tomas Germade[1], Ariane Bergmann[2], Robbie Loewith[2], Uwe Sauer[1], Duncan Holbrook-Smith[1]*

[1]Institute of Molecular Systems Biology, ETH Zurich, Zurich, Switzerland; [2]Department of Molecular Biology, University of Geneva, Geneva, Switzerland

**Abstract** Although the genetic code of the yeast *Saccharomyces cerevisiae* was sequenced 25 years ago, the characterization of the roles of genes within it is far from complete. The lack of a complete mapping of functions to genes hampers systematic understanding of the biology of the cell. The advent of high-throughput metabolomics offers a unique approach to uncovering gene function with an attractive combination of cost, robustness, and breadth of applicability. Here, we used flow-injection time-of-flight mass spectrometry to dynamically profile the metabolome of 164 loss-of-function mutants in TOR and receptor or receptor-like genes under a time course of rapamycin treatment, generating a dataset with >7000 metabolomics measurements. In order to provide a resource to the broader community, those data are made available for browsing through an interactive data visualization app hosted at https://rapamycin-yeast.ethz.ch. We demonstrate that dynamic metabolite responses to rapamycin are more informative than steady-state responses when recovering known regulators of TOR signaling, as well as identifying new ones. Deletion of a subset of the novel genes causes phenotypes and proteome responses to rapamycin that further implicate them in TOR signaling. We found that one of these genes, *CFF1*, was connected to the regulation of pyrimidine biosynthesis through URA10. These results demonstrate the efficacy of the approach for flagging novel potential TOR signaling-related genes and highlight the utility of dynamic perturbations when using functional metabolomics to deliver biological insight.

*For correspondence: hduncan@ethz.ch

Competing interest: The authors declare that no competing interests exist.

## Editor's evaluation

This work measures time-resolved metabolomes of 164 yeast mutants using a high-throughput method (FIA-MS). The dynamic, nontargeted measurements allow for an improved inference of gene function in the TOR pathway after a rapamycin treatment, including the annotation of three new genes in the TOR signaling pathway. This case study opens an avenue for combined studies of functional genetics and metabolism.

## Introduction

Despite the long-standing sequencing of the *Saccharomyces cerevisiae* genetic code (*Goffeau et al., 1996*), the characterization of the roles of genes and proteins within it is an ongoing process (*Wood et al., 2019*). Any systematic understanding of the cell will require a full mapping between genes and functions. Various approaches have been used to explore gene function at a genome-wide scale. Many of these approaches rely only on tracking changes in the fitness of mutants in different conditions (*Giaever et al., 1999*). Similarly, synthetic genome arrays can be used to explore gene function by identifying genetic interactions between genes of unknown function and genes whose functions are better characterized (*Costanzo et al., 2010*). These approaches have been very successful, but generally use fitness as a readout to infer function through a guilt-by-association approach (*Aravind,*

*2000*) and thus are limited to genes that influence fitness under the chosen experimental conditions. Other techniques have been developed where the effects of mutations on the cell can be tracked beyond fitness. These approaches include transcriptional profiling (*Velculescu et al., 1997*) and high content imaging (*Carpenter, 2007*), among others. There is enormous variability for these approaches in terms of the number of unique features that can be collected as well as the time and cost required per sample analyzed. These approaches can also reveal changes within the cell that are measurable but that would not translate into differences in fitness. With the advent of high-throughput metabolomics, including flow-injection analysis mass spectrometry (FIA-MS), it is possible to measure the metabolome profile of cells in less than a minute per sample (*Fuhrer et al., 2011*). Increased throughput in metabolomics has allowed the union of functional genomics and metabolomics, opening a new approach to the characterization of gene function at a genome-wide scale (*Fuhrer et al., 2017*).

The relationship between genetics and the metabolome of *S. cerevisiae* has been a long-standing area of inquiry (*Breunig et al., 2014*), and metabolomics has been used to understand gene function. For example, metabolomics has been used to uncover phenotypes for silent mutations (*Raamsdonk et al., 2001*). In the past, pioneering work has demonstrated that high-throughput approaches can be used on a genome-wide scale to measure amounts of amino acids in yeast deletion mutants for the purpose of characterizing regulatory principles of biosynthesis (*Mülleder et al., 2016a*). Other work in *Escherichia coli* has explored the effect of loss-of-function mutations on the metabolome at a genome-wide scale (*Fuhrer et al., 2017*). However, these genome-scale approaches are insufficient to fully associate genes and functions since not all genetic deletions will exert a measurable effect on the cell in every condition (*Giaever et al., 2002*). This may in part be due to limited metabolome coverage, but it may also be because these studies were performed under steady-state conditions. Under these conditions, cells are not exposed to any dynamic perturbation, allowing them to buffer their metabolism in such a way as to obscure the effects of different mutations on the cell (*Jost and Weiner, 2015*). Dynamic cellular perturbation may reveal functional relationships between genes by evading some portion of compensating changes within the cellular system. Tracking the dynamics of the metabolome for large numbers of mutants has historically been unattainable due to throughput limitations, but with the order-of-magnitude increase in measurement speed offered by flow-injection such experiments are now feasible.

The ability to precisely follow changes in the levels of metabolites is particularly important when investigating the cellular systems that regulate metabolism and growth. The target of rapamycin (TOR) signaling system is a core regulator of these decisions around growth in eukaryotic cells (reviewed in *González and Hall, 2017*). Many core components of TOR signaling have been elucidated (*Loewith et al., 2002*). This includes the discovery of two functionally distinct sets of TOR protein complexes, one of which (TORC1) is sensitive to inhibition by rapamycin and a second complex (TORC2) which is not (reviewed in *Eltschinger and Loewith, 2016*). Key questions remain regarding the roles of different TOR signaling genes in the regulation of metabolism. For example, although it has been shown that leucine can regulate TOR through interactions with its transporter SLC7A5 (reviewed in *Taylor, 2014*), the broader question of how other amino acid levels are able to regulate TOR is currently not fully characterized (*González and Hall, 2017*). Genome-scale investigations have been devised to identify genes that convey resistance or susceptibility to the TORC1 inhibitor rapamycin in different mutant forms (*Butcher et al., 2006*). Although these approaches have had successes, they do not provide specific information regarding how any given mutation affects the cell. By contrast, high-throughput metabolomics can provide specific insights by identifying which mutants lead to defects in different dimensions of the metabolic response of the cell to rapamycin while also allowing for guilt-by-association analysis on the basis of metabolome similarity.

In this work, we exploit dynamic high-throughput metabolome profiling to measure the metabolome profiles of 164 loss-of-function mutants in yeast, and newly associate three genes with TOR signaling. Further investigation of these mutants showed that they have proteomic alterations that further implicate them in TOR signaling. A subset of them also showed altered growth responses during nutrient upshifts where TOR signaling is important. One of these genes, *CFF1* (YML079W), is a gene of unknown function with structural similarity to auxin binding proteins in plants (*Zhou et al., 2005*) that has recently been shown to be required for the production of quorum-sensing compounds (*Valastyan et al., 2021*). We discovered that a *CFF1* loss-of-function mutant also shows altered pyrimidine metabolism during nutrient upshifts, likely due to altered expression of the pyrimidine

biosynthetic enzyme URA10. These results demonstrate that *CFF1* mutation alters the cell's response to rapamycin and nutritional shifts, and thus implicates it in TOR signaling.

## Results
### TOR mutants show altered metabolome responses to rapamycin treatment

We sought to establish the metabolome responses of different TOR-related mutants to rapamycin so that we could use those responses as a baseline to explore gene function for mutants not yet known to be involved in TOR signaling. Thus, we characterized the effects of a collection of 85 mutants in TOR-related signaling genes on the metabolome response of the cell to rapamycin (*Figure 1A*, *Supplementary file 1*). The mutant collection included deletions in regulators that act upstream of TORC1, as well as downstream kinases or genes involved TOR-related processes such as autophagy. Only nonessential mutants were selected for the collection. These TOR signaling mutants were cultivated on synthetic defined media with glucose and ammonium as sole carbon and nitrogen sources. Cultures were grown to an optical density at 600 nm ($OD_{600}$) of 0.7 at which point metabolites were extracted and a rapamycin treatment (400 ng/mL) was performed. Rapamycin treatment inhibits TORC1 and elicits important dynamic responses in terms of metabolite levels and downstream signaling events (*Stracka et al., 2014*). We aimed to capture these dynamic effects in our mutant strains by preparing polar metabolite extracts after 5, 30, 60, and 90 min of rapamycin treatment. These extracts were measured using FIA-MS (*Fuhrer et al., 2011*), a chromatography-free method that allows for the measurement of relative metabolite levels with a broad coverage of metabolite classes in less than a minute per sample. The effect of rapamycin treatment on wild-type yeast as measured by high-throughput FIA-MS was orthogonally validated by a longer liquid chromatography-mass spectrometry (LC-MS)-based measurement method (*Figure 1—figure supplement 1*). The similar metabolome response seen using both methods demonstrated that our high-throughput measurements were of sufficient quality to explore the relative differences in metabolome responses between the mutants in this study while providing the throughput required to study the large number of mutants.

TORC1 is known to regulate many metabolic functions, but amino acid and nucleotide metabolism exhibit characteristic changes when TORC1 is inactivated (*Oliveira et al., 2015*; *Xu et al., 2013*). Specifically, the levels of most amino acids increase after rapamycin treatment due to inhibition of translational initiation (*Berset et al., 1998*), with some exceptions such as serine (*Mülleder et al., 2016a*; *Oliveira et al., 2015*). Upon TORC1 inactivation through starvation, nucleotide degradation increases, which leads to increased pool sizes of nucleoside-related compounds (*Xu et al., 2013*). Since the yeast were cultivated in media containing glucose as a carbon source and ammonium as a nitrogen source, basal TORC1 signaling is high in these exponentially growing wild-type yeast until rapamycin is added to inhibit TORC1. Consistent with reports in the literature (*Oliveira et al., 2015*), changes in metabolite levels such as nucleosides were seen when wild-type yeast were treated with rapamycin (*Figure 1B*).

The TOR-related mutants within this study are drawn from a range of classes, with multiple genes involved in autophagy, positive and negative regulation of TOR signaling, signaling genes that are mechanistically downstream of TOR, and other classes (*Figure 1C*). Significantly altered metabolite levels (absolute log2 transformed fold-changes of >0.5 compared with wild-type with a p-value of <0.05) could be observed for mutants across all functional classes compared with wild-type (*Figure 1D*). The highest number of changes in metabolite levels across all mutants were observed after 90 min of rapamycin treatment (*Figure 1D*). These results show that treating the cells with rapamycin can reveal metabolic responses that are not measurable prior to treatment.

Deleting genes that are positive upstream regulators of TOR signaling reduces the signal propagated through that system (*Péli-Gulli et al., 2015*), and thus either causes changes in the metabolome that resemble those resulting from rapamycin treatment (*Mülleder et al., 2016a*) or could further sensitize the cell to TORC1 inhibition. *gtr1*, a mutant in a key positive regulator of TOR signaling that acts upstream of TORC1 (*Binda et al., 2009*), showed elevated levels of glutamine compared with wild-type in the absence of rapamycin (*Figure 1E*). Since glutamine levels increase after rapamycin inhibition of TORC1, this increase in glutamine in *gtr1* serves as a positive control. Phenylalanine also shows an accumulation after cells were treated with rapamycin, but it showed similar levels in

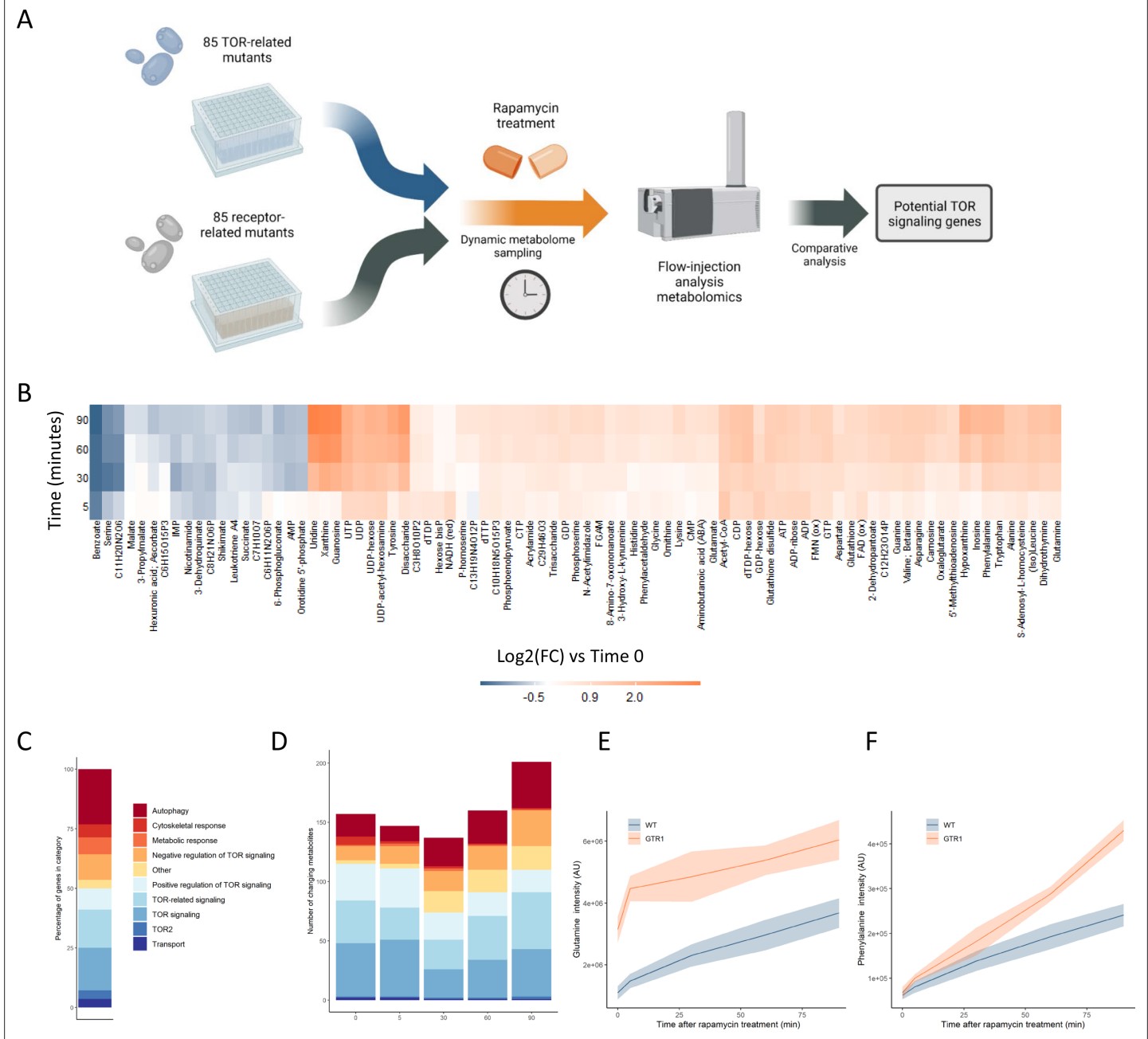

**Figure 1.** Metabolome profiling of TOR and receptor-related mutants captures dynamic metabolome changes. (**A**) A schematic diagram illustrating the study design. (**B**) The effect of rapamycin treatment over time is shown for wild-type cells after the indicated duration of rapamycin treatment compared with the untreated control condition. Only metabolites that show a significant (p-value<0.05, two-sided *t*-test) are shown. (**C**) The share of TOR-related mutants included in the study falling into each of the indicated categories is indicated. (**D**) The number of significantly changing metabolites across the mutants in the indicated categories is shown for each duration of rapamycin treatment. Significantly changed metabolites show an absolute log2 transformed fold-change compared with wild-type of >0.5, and a p-value of <0.05 based on a two-sided Student's *t*-test. (**E, F**) The normalized ion intensities for glutamine and phenylalanine in wild-type and *gtr1* yeast are shown after the indicated duration of rapamycin treatment. The central lines indicate the median value across four biological replicates. Shaded areas indicate the standard deviation across replicates.

The online version of this article includes the following figure supplement(s) for figure 1:

**Figure supplement 1.** Comparison of effects of rapamycin treatment on the metabolome as measured by flow injection and liquid chromatography-mass spectrometry (LC-MS).

**Figure supplement 2.** Positive regulators of TORC1 signaling show similar metabolite changes.

**Figure supplement 3.** The effect of rapamycin on the levels of metabolites in wild-type and *atg13* yeast.

**Figure supplement 4.** The effect of normalization of flow-injection metabolomics data.

untreated wild-type and *gtr1* cells (*Figure 1F*). However, the *gtr1* mutant showed a much stronger accumulation of the amino acid than did wild-type after rapamycin treatment (*Figure 1F*). Thus, some metabolic alterations manifested by deleting genes involved in TOR signaling are only revealed after treatment with rapamycin. When this analysis was expanded to all deletion mutants in genes that code for positive regulators of TORC1 signaling within our collection (*GTR1* [*Binda et al., 2009*], *GTR2* [*Binda et al., 2009*], *LST4* [*Péli-Gulli et al., 2015*], *MTC5* [*Panchaud et al., 2013*], *RTC1* [*Panchaud et al., 2013*], *SEA4* [*Panchaud et al., 2013*]; hereafter referred to as 'positive regulators'), similar effects were seen across mutants and time points (*Figure 1—figure supplement 2*). These results suggest that metabolome profiles could be used to search for novel positive regulators of TORC1 signaling.

## Metabolome profiling identifies novel potential TORC1 signaling genes

Building on the investigation of known TOR signaling genes, we assembled another collection of 85 loss-of-function mutants. These mutants were selected for being known receptors, nutrient sensing proteins in *S. cerevisiae*, showing sequence similarity to known receptors in other species, or carrying protein domains that are common in receptor proteins (*Supplementary file 1*). Mutants from this collection are from hereon referred to as 'receptor-related.' The collection includes both intracellular and extracellular receptors. By enriching the mutant collection for receptor-related functions, we aimed to increase our chances of identifying genes that are involved in the regulation of TORC1 signaling. Dynamic metabolome extracts for these 85 mutants (from here on referred to as receptor-related) were collected under the same conditions as described above for the TOR mutants (*Figure 1A*) and were analyzed by FIA-MS.

Metabolome profiles for each mutant at each time point were determined by calculating the average log2 fold-change between the metabolite intensities of the mutants compared with the wild-type control. Patterns of metabolome similarity across the dataset were systematically analyzed by calculating the Pearson correlation between the metabolome profiles for each mutant in the TOR and receptor-related collections to each other for each time point. Manhattan distances between each mutant's profiles of correlation were then calculated and the mutants were subjected to hierarchical clustering (*Figure 2A*). This analysis showed clustering of the positive regulators, with most co-clustering mutants being other TOR-related genes. Indeed, the empirical likelihood for the distances between positive regulators being as low as was observed was calculated to be <5% for all time points that were tested (*Figure 2B*). The median distance between each mutant and the other positive regulators was used as a metric for the binary classification of mutants as being positive regulators or not. Annotated positive regulators of TORC1 signaling were treated as true positives, and all other mutants in the annotated collection were treated as false positives. When the area under a receiver operating characteristic curve was calculated for each time point, the values were >0.5 for all the time points (*Figure 2BC*). The area under the curve for the untreated cells was observed to be the smallest of all the time points, indicating that a dynamic perturbation by rapamycin treatment improved the recovery of true positives (*Figure 2B*). This highlights the utility of dynamic perturbations when applying metabolomics to studying gene function. Previous studies have interrogated gene function by measuring the genetic interactions between mutations in yeast systematically, including for mutations in genes included in this study. It is possible that the metabolome-based distances that were calculated here capture the same information that arises from genetic interaction studies. However, when the metabolome distances between mutants were compared with the genetic interaction score between genes (*Costanzo et al., 2016*), small but significant correlations were seen between the datasets (*Figure 2—figure supplement 1*). These correlations were largely driven by relatively few interactions with large negative genetic interaction scores. This demonstrates that, with the exception of very strong negative genetic interactions, metabolome-based distances between mutants provide additional descriptions of the functional relationships between mutants. This suggests that new patterns of similarity derived from metabolomics could be used to identify new functional relationships.

This analysis captures other expected relationships between the genes within the dataset. For one, TOR-related genes showed smaller distances to the positive regulators compared with the receptor-related genes (*Figure 2A*). This is expected since although these genes are not upstream positive regulators of TORC1 signaling, many of them are effectors of TORC1 signaling that are involved in the

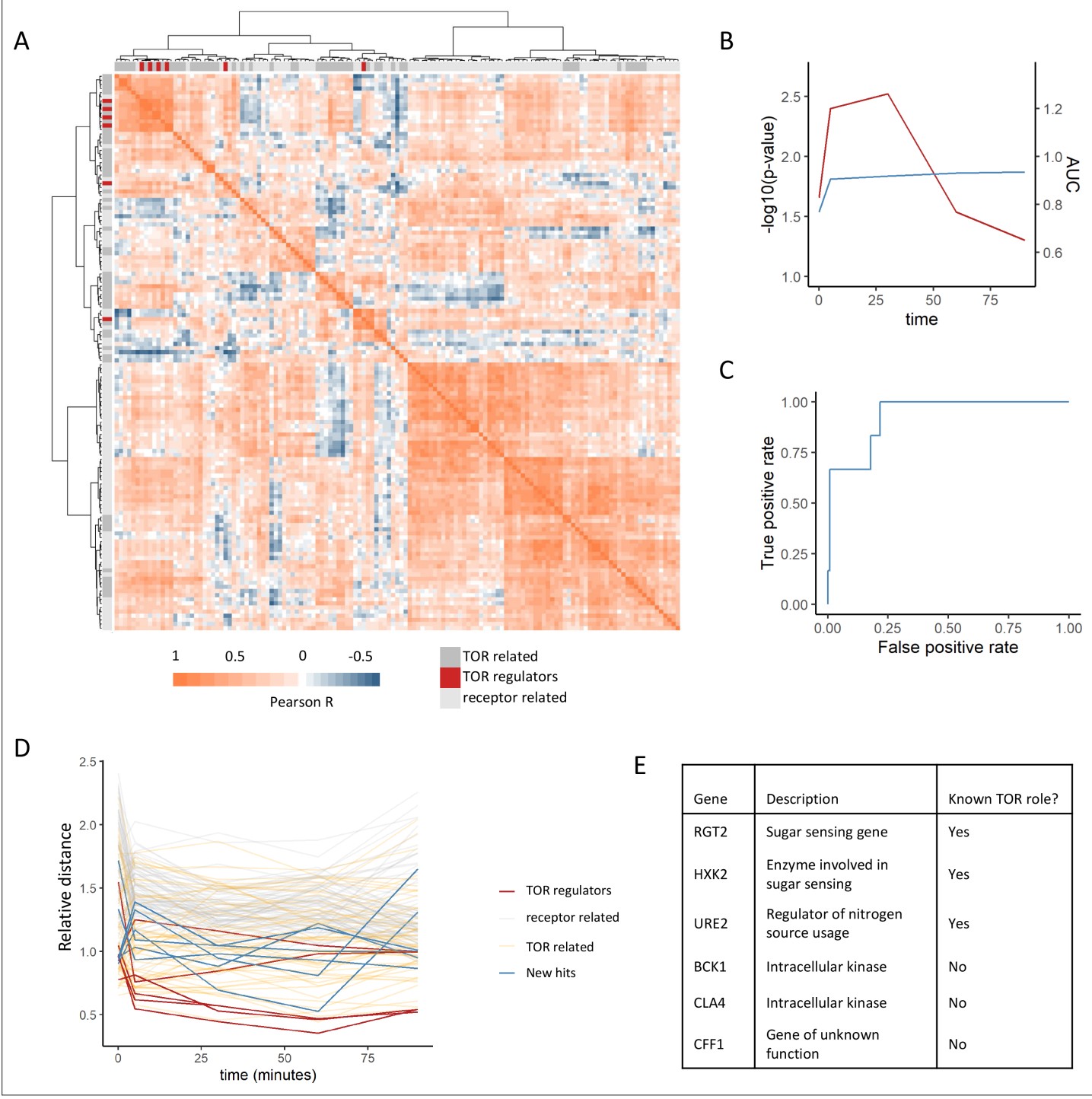

**Figure 2.** Positive regulators of TOR signaling cluster together based on their metabolome profiles after rapamycin treatment. (**A**) The Pearson correlation matrix between all TOR and receptor-related genes is shown for the 30 min rapamycin treatment condition with clustering based on Ward's algorithm of the Manhattan distance. The identity of the clustered mutants (TOR-related, upstream positive regulators of TOR signaling, or receptor-related) is indicated with the alternate colored heatmap as the edge of the correlation heatmap. (**B**) The area under the curve for the recovery of known upstream positive regulators of TOR signaling is shown for each time point at which metabolomics data was collected. Area under the curve (AUC) is indicated as a blue line. The empirical p-value for the median distance between those genes is also shown for each time point, based on data randomization. The p-value is indicated as a red line. (**C**) A receiver operating characteristic curve is shown for the 30 min time point demonstrating the recovery of known positive recovery of known upstream positive regulators of TOR signaling. (**D**) The distance for each mutant relative to the 0.2 false discovery rate (FDR) threshold calculated for the time point is shown across all time points and genes in this study. Values of <1 indicate that the mutant

*Figure 2 continued on next page*

*Figure 2 continued*

has passed the distance threshold for that time. (**E**) Table describing the identity and role of receptor-relating genes that pass the 0.2 FDR threshold for distance from positive regulators of TOR signaling.

The online version of this article includes the following figure supplement(s) for figure 2:

**Figure supplement 1.** Relationship between metabolome distance and genetic interaction score.

**Figure supplement 2.** The metabolome distances between positive and negative regulators of TOR signaling.

cellular response to altered TORC1 signaling and thus should share some features of their metabolic response. Additionally, genes that are negative regulators of TORC1 signaling (*NPR3* [*Neklesa and Davis, 2009*], *PBP1* [*Yang et al., 2019*], *PSR1* [*Chen et al., 2018*], *PSR2* [*Chen et al., 2018*], *TIP41* [*Jacinto et al., 2001*], and *WHI2* [*Chen et al., 2018*]) showed among the longest distances to the positive regulators within the dataset (*Figure 2—figure supplement 2*). This is expected since the metabolome response of a loss-of-function mutant in a negative regulator of TORC1 would be quite distant from those of positive regulators. This, therefore provides additional evidence that metabolome-based distances can capture functional relationships between genes.

Given that our analysis was able to recover known positive regulators of TORC1, we extended this analysis to ask which receptor-related mutant metabolome profiles were consistent with being a positive regulator of TORC1. For each time point, the distance at which a false-positive rate of 0.2 was obtained for the recovery of positive regulators was determined. Six receptor-related mutants (*RGT2, HXK2, URE2, BCK1, CLA4,* and *CFF1*) were able to pass this cutoff and were selected as potential novel positive regulators (*Figure 2DE*). Out of these genes, three are involved in signaling pathways with known cross-talk with TORC1 signaling. Namely, *RGT2* and *HXK2* are involved in sugar sensing via the PKA signaling, which feeds into the regulation of downstream TORC1 signaling targets (*Shashkova et al., 2015*). *URE2* is a transcriptional regulator for nitrogen catabolite repression, which is in part regulated by TORC1 (*Shashkova et al., 2015*). The other hits included the intracellular kinases *BCK1* and *CLA4*, as well as a gene of poorly characterized function *YML079W*, which has recently been named *CFF1* based on its inability to produce the compound 4-hydroxymethylfuranone (*Valastyan et al., 2021*). These results show that a metabolome profiling-based guilt-by-association approach can be used to identify genes with known cross-talk with TOR signaling as well as genes whose connections to TOR signaling are completely novel.

Beyond the guilt-by-association approach outlined above, the data presented here can also be viewed at the level of the individual mutant and metabolite. This allows us to assess the involvement of different genes in specific metabolic processes. This stands in contrast to other guilt-by-association approaches that are based solely on fitness and cannot directly observe the effect of mutants on metabolic processes. One of the strongest examples is *atg13*, which is incapable of performing autophagy (*Funakoshi et al., 1997*). This mutant shows a strong reduction in the accumulation of the nucleoside inosine under rapamycin treatment (*Figure 1—figure supplement 3A*). This is likely caused by the inability of the mutant to perform autophagy under these conditions (*Xu et al., 2013*), but other metabolites such as glutamine show no difference in their accumulation (*Figure 1—figure supplement 3B*) or in the case of kynurenine a more limited effect is observed (*Figure 1—figure supplement 3C*). These results highlight a clear relationship between *ATG13* and nucleoside pool sizes under rapamycin treatment that likely results from the disruption of autophagy, but also raises questions about the mechanism through which *ATG13* affects the levels of metabolites like kynurine. This is an example of the type of observation that can be obtained from this analysis that would be missing from a more traditional guilt-by-association approach. Although *ATG13* deletion had one of the strongest effects on metabolite levels within the dataset, differences in the metabolome profiles of many genes across a range of different functional classes were observed in this study (*Figure 1D*). Thus, the data included within this study can be used to query the effect of the deletion of genes of interest on the metabolome. The response of the cell to rapamycin is not restricted to changes at the level of the metabolome, so we set forth to investigate the proteome response of the mutants in the newly predicted positive regulators of TORC1 signaling to rapamycin.

## New and predicted TOR signaling genes show altered proteome responses to rapamycin

Metabolome-based guilt-by-association was used to newly implicate *CFF1*, *BCK1*, and *CLA4* in TORC1 signaling. It is hypothesized that if these genes are involved in TORC1 signaling, the effect of their deletion on the cell is mediated at least in part by changing protein levels. We would also expect these changes to be similar to those seen in mutants in known positive regulators of TOR signaling. We tested this by analyzing the proteomes of six mutants in known positive regulators of TORC1 signaling, and the six receptor-related mutants that were associated with positive regulation of TORC1 signaling as was described above (*Figure 2E*). Cultures were grown to an $OD_{600}$ of approximately 0.8 in defined media with glucose and ammonium as carbon and nitrogen sources. The strains were treated with 400 ng/mL rapamycin and were then grown for 30 min before harvesting the yeast and subjecting them to label-free, quantitative proteomics (*Demichev et al., 2020*). Treatment of wild-type yeast with rapamycin resulted in broad changes in the proteome, with enriched changes in the levels of proteins in gene ontology (GO) biological process categories such as carbohydrate and organic acid metabolism (*Figure 3A*), recapitulating patterns observed in previously published work (*Iesmantavicius et al., 2014*). TOR signaling gene deletion mutants would be expected to show altered proteome responses to rapamycin treatment compared with the proteome responses seen in wild-type. The proteins that changed in abundance upon rapamycin treatment in all 12 mutants were tested for enrichment of the GO terms that were enriched for wild-type treated with rapamycin. All mutants, except HXK2 and RGT2, showed clearly altered patterns of GO enrichment upon rapamycin treatment (*Figure 3A*), as would be expected for genes that are required to respond to rapamycin.

In addition to an altered proteome response to rapamycin, BCK1, CLA4, and CFF1 also showed similar proteomic profiles to known positive regulators of TOR signaling (*Figure 3B*). The relative changes in protein levels were calculated for each mutant compared with wild-type. The Pearson correlation between those mutant proteome profiles was then calculated and used to assess how similar the effects of the mutations were on the proteome. The correlations coefficients were universally positive, with values as high as 0.91 (*Figure 3B*). Some mutants with similar molecular functions were seen to cluster together (both HXK2 and RGT2 are involved in sugar sensing) but *cla4*, *bck1*, *cff1*, and *ure2* proteomes clustered with the known positive regulators of TORC1 signaling. Since yeast were grown in conditions where TORC1 signaling is active, a mutant that reduced TORC1 signaling would cause a change in the proteome that is similar to that of rapamycin-treated cells. All tested mutants showed a positive correlation in their proteome compared with that of wild-type treated with rapamycin, but *cff1* and *bck1* showed the strongest correlations with rapamycin treatment. This shows that the changes in the proteome that were caused by deletion have similar effects on the proteome to reduction of TORC1 signaling, as would be expected if those genes play a positive role in TOR signaling.

## Newly predicted TORC1-related mutants show TORC1-related phenotypes

The above results implicate six receptor-related genes in the positive regulation of TORC1 signaling. TORC1 signaling is central to the adaptation of the yeast cell to changing nutritional environments. Therefore, mutations that impair TORC1 signaling reduce the ability of the cell to adapt to increases or decreases in nutritional quality (*Kira et al., 2014*). We tested whether deletion mutants in *CFF1*, *RGT2*, *HXK2*, *URE2*, *CLA4*, and *BCK1* showed such an impairment by performing nutritional upshift experiments. Cultures were grown in minimal media with proline as a nitrogen source and glucose as a carbon source. Under these conditions, TORC1 activity is reduced due to the poor nitrogen source quality (*Stracka et al., 2014*). After cultivation for 18 hr, the cultures were exposed to a nutritional upshift through the introduction of ammonium sulfate into the cultures, after which the growth rate and lag time were determined for all the predicted positive regulator mutants. Under these conditions, TORC1 signaling will become more active than before the addition of the ammonium sulfate, and genes involved in the positive regulation of TORC1 signaling would be expected to be physiologically relevant. *cff1* and *bck1* showed significantly reduced growth rates upon the supplementation of the media with ammonium sulfate (*Figure 3C*), and *hxk2* demonstrated a longer lag time (*Figure 3D*). Previously published functional genomics screens have shown that known positive regulators of TORC1 signaling show reduced growth rates when grown in minimal media compared

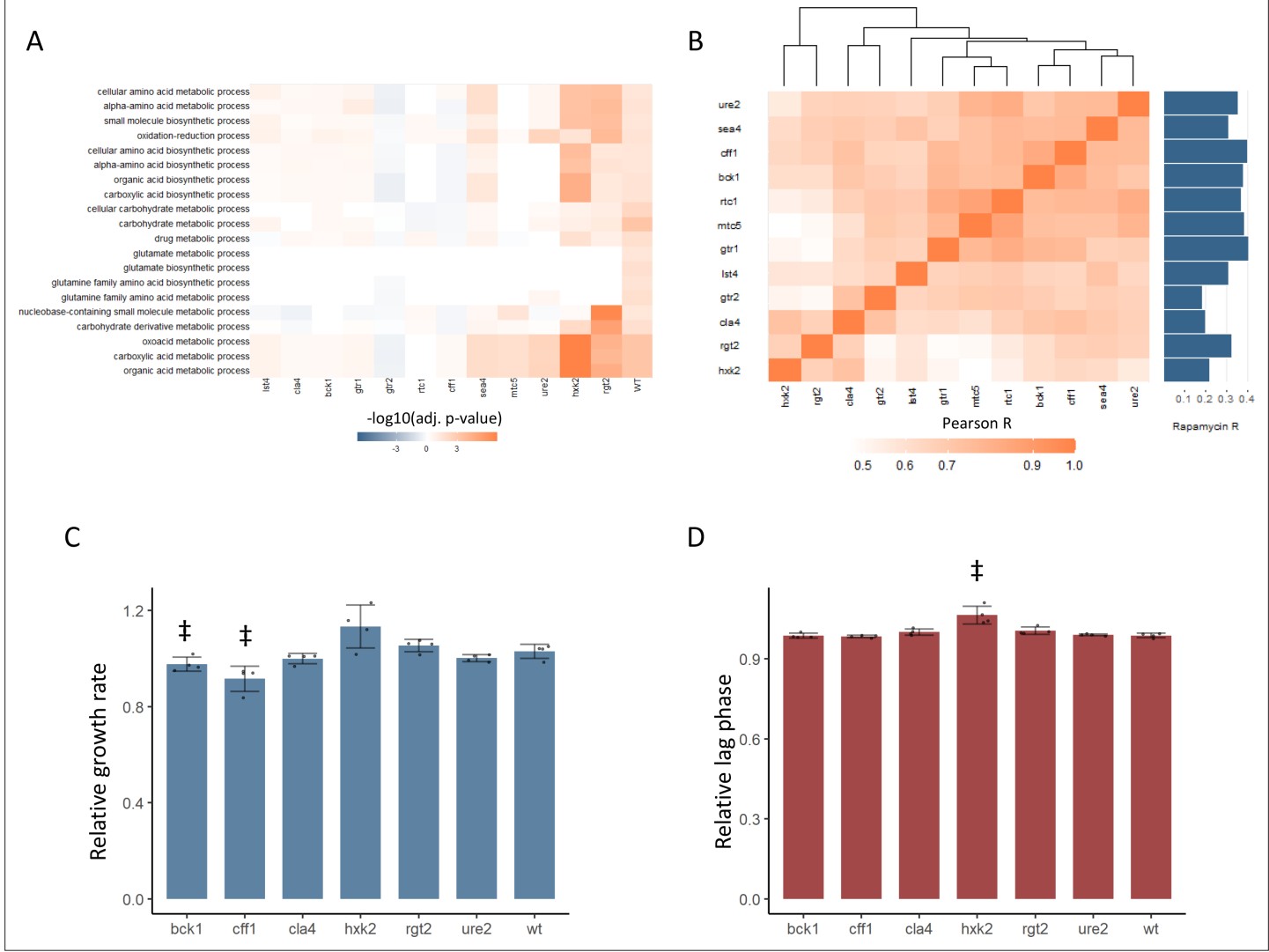

**Figure 3.** New TOR hits resemble positive TORC1 regulators at the level of the proteome and growth. (**A**) -log10 transformed adjusted p-values are shown for the enrichment of gene ontology (GO) biological processes in changing proteins after rapamycin treatment in each mutant. Only GO terms that show enrichment in wild-type yeast treated with rapamycin are included within the plot. Adjusted p-values are Benjamini–Hochberg corrected hypergeometric test outcomes. Negative values indicate an enrichment among proteins of decreased abundance. (**B**) The Pearson correlations between the protein fold-changes observed when comparing each mutant to wild-type are shown. The Pearson correlation of each mutant proteome to the proteome response of wild-type treated with rapamycin is also shown in bar format to the right of the plot. (**C**) The maximum relative growth rate for the indicated mutants and wild-type is shown after an upshift from using proline as a nitrogen source to ammonium sulfate. The bar height indicates the mean value, and error bars represent the standard error of the mean from four biological replicates; individual points indicate the average value from one biological replicate. ‡p-value<0.05 for a two-sided *t*-test compared with the wild-type control. (**D**) The lag time for the indicated mutants and wild-type is shown after an upshift from using proline as a nitrogen source to ammonium sulfate. The bar height indicates the mean value, and error bars represent the standard error of the mean from four biological replicates; individual points indicate the average value from one biological replicate. ‡p-value<0.05 for a two-sided *t*-test compared ith the wild-type control.

with wild-type (*Breslow et al., 2008*). In that study, the relative growth rates of the positive regulators varied from 0.71 to 0.97 with an average value of 0.84 for the five positive regulators that were measured (*Breslow et al., 2008*). This means that the reduction of growth rate seen for *cff1* and *bck1* was within a similar range as was reported previously for the known positive regulators as described above. Although *CFF1* and *BCK1* have not previously been considered TORC1 signaling genes, these mutants showed reduced growth rates during a nitrogen source upshift as would be expected for a mutation in a positive regulator of TORC1 signaling. These results, in addition to their metabolome and proteome level similarity to known positive regulators of TORC1 signaling, further implicate these

two genes in the TORC1 signaling. Although *CFF1, BCK1,* and *CLA4* all show signs of involvement in TORC1 signaling, because *CFF1* is a protein of unknown function, we decided to investigate its possible role in TORC1 signaling in greater depth.

The protein SCH9 is directly phosphorylated by TORC1 and is a key node for transmission of signaling intro processes downstream of TORC1 (*Urban et al., 2007*). If CFF1 acts mechanistically downstream of TORC1 and SCH9, we would expect that the mutant would not impact the ability of TORC1 to phosphorylate SCH9 under nutrient-rich conditions. As expected, Western blot quantification revealed a similar degree of SCH9 phosphorylation in wild-type and *cff1* strains in rich conditions (*Figure 4A*). Since CFF1 is required for a normal response to changing nutritional environments, it therefore appears to act mechanistically downstream of TORC1. Consistent with this expectation, the relative abundance of the CFF1 protein was reduced upon treatment rapamycin (*Figure 4B*).

If CFF1 is involved in TORC1 signaling, it should also play a role in the regulation of the cell state under nutritional downshifts. To this end, wild-type and *cff1* yeast were cultivated in minimal media with ammonium as the sole nitrogen source and then shifted into media where the poor nitrogen source proline, 30 min prior to metabolite extraction. Under these conditions, TORC1 should shift from an activated to an inactive state. If *CFF1* is required to adapt to the poor nitrogen source, we would expect to see differences in metabolite pools for the mutant compared with the wild-type. LC-MS analysis of the samples indicated that pyrimidine biosynthetic precursors (carbamoyl aspartate and dihydroorotate) were increased in *cff1* compared with wild-type (*Figure 4C*). TOR signaling has been shown to stimulate de novo pyrimidine biosynthesis in other systems (*Ben-Sahra et al., 2013*; *Robitaille et al., 2013*), so the observation that *CFF1* is required for the maintenance of pyrimidine precursor pool sizes during this nitrogen source shift further implicates its TORC1-related metabolic processes.

Since *CFF1* deletion increased the amounts of pyrimidine precursors under a nitrogen downshift, we queried the proteomic data to determine whether these changes could be explained by alterations in the levels of enzymes involved in that pathway. The enzyme URA10, which converts orotate into oratidine-5-phosphate (*Figure 4D*), showed a significant decrease in abundance (*Figure 4E*), which was not shared by the other members of the metabolic pathway (*Figure 4—figure supplement 1*). This alteration of the expression of URA10 provides a likely explanation for the effect of *CFF1* deletion on the abundances of metabolites upstream of that enzyme in the pyrimidine biosynthetic pathway. Taken together with earlier results, CFF1 is required for normal adaptation of the cell to changing metabolic environments in terms of both up- and downshifts with a specific role in the regulation of pyrimidine biosynthesis through URA10. This is in addition to its similarity to positive regulators of TORC1 signaling at the level of metabolome and proteome. This offers an example of how gene function can be explored using high-throughput metabolomics using a guilt-by-association approach during a dynamic perturbation.

## Discussion

In this study, we used high-throughput metabolomics in a guilt-by-association framework to identify mutants with metabolome responses to rapamycin that are similar to those of mutants in known positive regulators of TOR signaling. Using this approach, we were able to recover known genes involved in positive regulation of yeast TOR signaling based on the relationships between the metabolome profiles of the mutants. Notably, the recovery of known positive regulators of TORC1 signaling was highest after the cells were dynamically perturbed with rapamycin. To our knowledge, this is the first demonstration that dynamic perturbation of the cell improves the recovery of eukaryotic gene function when using a metabolomics-based guilt-by-association scheme. These studies also provide insights into why different genome-wide metabolome profiling experiments conducted under steady state appear to provide incomplete information regarding gene function (*Mülleder et al., 2016a*; *Fuhrer et al., 2017*).

Through our guilt-by-association approach, we were able to use patterns of metabolome similarity to recall known positive regulators of TORC1 signaling and implicate six receptor-related genes in that process as well. Three of the recovered receptor-related genes (*HXK2, RGT2,* and *URE2*) are known to have connections to TOR signaling. *CFF1, BCK1,* and *CLA4* were newly predicted to be involved in the positive regulation of TORC1 signaling. *CFF1* is of particular interest because it is a gene of unknown function, its mutation has previously shown phenotypes including a reduced resistance

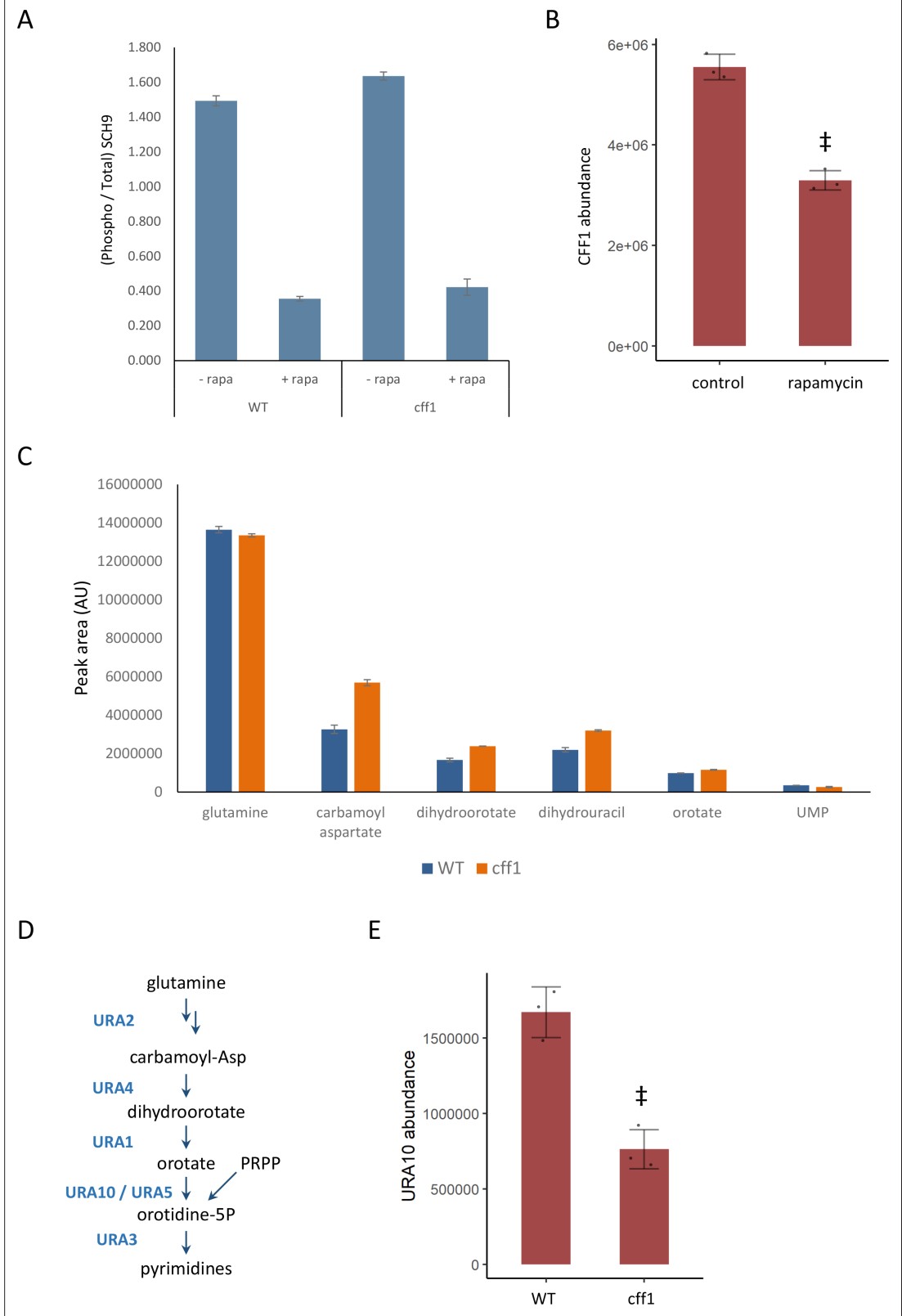

**Figure 4.** CFF1 acts downstream of TORC1 and in the regulation of pyrimidine metabolism. (**A**) The relative ratio of phosphorylated SCH9 to total SCH9 is shown for wild-type and *cff1* yeast grown in rich conditions before and after a 30 min treatment with 400 ng/mL rapamycin. Error bars indicate the standard deviation of biological replicates (n = 3). (**B**) The average abundance of CFF1 is shown in wild-type yeast either treated with rapamycin (400 ng/mL) or a control. Points represent a single replicate, bar height indicates the average value, and the error bar indicates the standard deviation (n = 3

*Figure 4 continued on next page*

*Figure 4 continued*

biological replicates). ‡p-value<0.05 for a two-sided *t*-test compared with the control. (**C**) The peak areas for the indicated metabolites are indicated for both WT and *cff1* yeast that were exposed to a nitrogen source downshift for a duration of 30 min. Data was collected by liquid chromatography-mass spectrometry (LC-MS) as indicated in the 'Methods' section. Bar heights indicate the average value for three biological replicates, with the error bars indicating the standard deviation. (**D**) A subset of the pathway for pyrimidine biosynthesis is diagrammed with enzymes colored blue and metabolites written in black. (**E**) The average abundance of URA10 is shown in *cff1* yeast or a wild-type control. Points represent a single replicate, bar height indicates the average value, and the error bar indicates the standard deviation (n = 3 biological replicates). ‡p-value<0.05 for a two-sided *t*-test compared ith the control.

The online version of this article includes the following figure supplement(s) for figure 4:

**Figure supplement 1.** The effect of rapamycin and *CFF1* deletion on the levels of pyrimidine biosynthetic enzymes.

**Figure supplement 2.** The effect of normalization of proteomics data.

to hyperosmotic stress (*Yoshikawa et al., 2009*), and reduced chronological life span (*Campos et al., 2018*). Previous work has also demonstrated structural similarity between CFF1 and auxin-binding proteins from plants (*Zhou et al., 2005*), as well as a role in determining the production of the compound 4-hydroxymethylfuranone (*Valastyan et al., 2021*). Further exploration of the *cff1* response to rapamycin at the proteome level revealed a clearly impaired response to rapamycin, a similarity between the effects of *cff1* on the proteome to those of rapamycin, and a reduction in CFF1 abundance under rapamycin treatment. We showed that *CFF1* deletion caused a defective response to nutrient upshifts at the level of growth and altered metabolite pool sizes and during a nitrogen quality downshift. The phosphorylation of the key TORC1 target SCH9 was not altered in *cff1* yeast, suggesting that the mechanism of action of the gene lies downstream of TORC1. Intriguingly, CFF1 has been reported to be phosphorylated at a number of positions, including serine 68 under glucose limitation (*Lanz et al., 2021*). This could point to additional regulation of CFF1 activity at a post-translational level within the context of TOR signaling downstream of TORC1. Although the exact mechanistic role of CFF1 in TORC1 signaling will require deeper investigation, CFF1 seems to act downstream of TORC1 to regulate pyrimidine biosynthesis through URA10. Unlike *CFF1*, both *BCK1* and *CLA4* have mammalian homologs. *BCK1* has homology to mammalian Map3K1, which is implicated in breast cancer in humans (*Easton et al., 2007*). PAK4, PAK5, and PAK6 are homologs of *CLA4*, and PAK6 has also been implicated in clear cell renal cell carcinoma (*Liu et al., 2014*). This suggests that investigating the relationships between these genes and TORC1 signaling could provide additional insights into cancer biology in humans, including novel therapeutic targets.

In addition to the guilt-by-association approach described above, our approach has the benefit of allowing for the exploration of the effects of mutants on the relative quantities of many metabolites under perturbation with rapamycin. This allowed us to capture the role of *ATG13* in the regulation of nucleoside levels in the cell under rapamycin treatment, but also raises questions about the mechanism by which it regulates the abundances of other metabolites under rapamycin treatment. These kinds of observations can be made across many other mutants that are included in this dataset and can be the basis of future work where the roles of genes in metabolic regulation can be explored by members of the scientific community. To enhance the usability of this data, we assembled an interactive data visualization app where users can browse through this data in order to investigate mutants that are of interest to them (https://rapamycin-yeast.ethz.ch). Furthermore, our data are made available in raw form, and because the acquisition of the main dataset was performed in an untargeted mode, these data can be reanalyzed in the future as the library of potential metabolites is expanded. This can allow for even greater utility for researchers with a particular interest in any compound that may be unannotated within our analysis. In addition to this metabolomics data, this article includes the proteome response of 12 mutants to rapamycin and thus provides a resource for members of the community who wish to further explore the proteome response of the cell to rapamycin treatment in those strains.

In this work, we demonstrate that condition-specific, dynamic metabolome profiling can offer attractive properties for the exploration of gene function compared to steady state metabolomics measurements. This builds on work showing that dynamic perturbations of gene expression can be used to recover drug-target relationships on the basis of metabolome similarity (*Holbrook-Smith et al., 2022*; *Anglada-Girotto et al., 2022*) and thus further demonstrates the value of non-steady-state perturbations of the cell in the context of metabolome profiling. Although this approach included

measurements of the metabolome at five time points, subsequent analysis at a genome-wide level could just as easily be performed with a single time point within the dynamic perturbation. Thus, we can reveal relationships between metabolites and mutants that could be hidden during steady state due to homeostatic changes in gene expression or other compensatory changes within the cell. This would allow for the discovery of hitherto uncharacterized relationships between genes as well as identify novel roles for genes in metabolic regulation.

## Methods

### Yeast cultivation

Liquid-cultivated yeast were grown at a temperature of 30°C with a shaking frequency of 250 rpm. Auxotrophic strains were cultivated in YPD medium (10 g/L yeast extract [BD Biosciences: 288630], 20 g/L Bacto-peptone [BD Biosciences: 211830], 5 g/L agar [BD Biosciences: 214530]). Yeast were transformed as described previously (*Gietz and Schiestl, 2007*) with the pHLUM plasmid (*Mülleder et al., 2016b*) and selected for growth on SD media with inositol 5 g/L ammonium sulfate (Sigma-Aldrich: A4418), 1.7 g/L Yeast Nitrogen base (BD Biosciences: 233530), 20 g/L D-(+)-glucose (Sigma-Aldrich: G8270), and 10 mg/L myo-inositol (Sigma-Aldrich: I5125). For growth shift experiments, yeast were cultivated in or shifted to SD proline media (5 g/L potassium sulfate [Sigma-Aldrich: P0772], 1.7 g/L Yeast Nitrogen base [BD Biosciences: 233530], 20 g/L D-(+)-glucose [Sigma-Aldrich: G8270], 750 mg/L proline [Sigma-Aldrich: P0380], and 10 mM potassium phthalate [pH 5, Sigma-Aldrich: 60360]).

### Yeast strains

Auxotrophic haploid deletion strains were recovered from the Euroscarf haploid mating type a library (*Taylor, 2014*) and were transformed as described above (*Gietz and Schiestl, 2007*) with the pHLUM plasmid (*Mülleder et al., 2016b*) in order to restore prototrophy.

### Metabolite extraction

Yeast were cultivated such that after at least two doublings metabolite extractions could be performed on cultures with an average $OD_{600}$ of approximately 1.0. The yeast were grown at 1.2 mL scale in 96-well plates with $OD_{600}$ measured intermittently throughout the sampling. For time-course rapamycin experiments, metabolites extracts were taken immediately before rapamycin treatment, and then at 5, 10, 30, 60, and 90 min after treatment with the drug at a final concentration of 400 ng/mL. Sampling was performed by harvesting 100 μL aliquots of yeast by centrifugation for 2 min at 2250 rcf in a 4°C precooled centrifuge, and discarding the supernatant by vigorous inversion. 100 μL of –2°C extraction solution (40% [v/v] HPLC-grade acetonitrile [Sigma-Aldrich: 34998], 40% [v/v] HPLC grade methanol [Sigma-Aldrich: 34885], 20% [v/v] HPLC-grade water [Sigma-Aldrich: 1153331000]) was added to the cell pellets, and extractions were allowed to proceed for 18 hr at –2°C. Extracts were then stored at –8°C in sealed conical plates (Huber lab: 7.0745, Huber lab: 7.1058).

### Flow-injection time-of-flight mass spectrometry for metabolomics

Flow-injection analysis for mass spectrometry-based metabolomics was performed using an Agilent 6550 Series quadrupole time-of-flight mass spectrometer (Agilent) by an adaptation of the method described by *Carpenter, 2007*. The analysis was performed utilizing an Agilent 1100 Series HPLC system (Agilent) coupled to a Gerstel MPS 3 autosampler (Gerstel). The mobile phase flow rate was set at 0.15 mL/min, with the isocratic phase composed of 60:40 (v/v) isopropanol and water buffered to a pH of 9 with 4 mM ammonium fluoride. The instrument was run in 4 GHz mode for maximum resolution while collecting mass spectra between 50 and 1000 m/z. Online mass axis correction was performed with taurocholic acid and Hexakis (1H, 1H, 3H-tetrafluoropropoxy–phosphazene) within the mobile phase.

### Flow-injection data analysis

Processing of mass spectra, including centroiding, merging, and ion annotation, was performed as described in *Carpenter, 2007*. Raw annotated ion intensities are provided in *Supplementary file 2*. Data was normalized and analyzed in Python using the Pandas package (*McKinney, 2019*). Datasets

were filtered for outliers in terms of the biomass at the time of sampling as well as in total ion current. Raw ion intensities were normalized to counteract temporal drifts, as well as $OD_{600}$ effects. In both cases, a locally weighted scatterplot smoothing (LOWESS) regression approach was used to remove trends in the data arising from those parameters. The effect of normalization on data quality is visualized in *Figure 1—figure supplement 4*. Average metabolite intensities were compared between each mutant with wild-type at each time point in order to calculate an average metabolome profile in the form of log2 fold-changes.

## Distance analysis of metabolome profiling

Within each time point, the Pearson correlation coefficients between all metabolite log2 fold-changes for each mutant compared with the wild-type control were calculated. Those correlations were clustered according to Ward's method applied to Manhattan distances between mutant correlation matrices. The average distance between each mutant and genes that are annotated as positive regulators of TORC1 signaling (GO 0032008) were then calculated. For known positive regulators of TORC1 signaling, the distance was calculated with them excluded from the list of known positive regulators of TORC1. Empirical p-values were determined through 10,000 randomizations of labels on correlation profiles in order to determine the distribution of distances between known positive TORC1 signaling genes.

## Liquid-chromatography mass spectrometry for metabolomics

Normal-phase chromatography was used to separate the metabolite extracts. Chromatographic separation was performed using an Agilent Infinity 1290 UHPLC stack with Agilent 1100 Series binary pump with an InfinityLab Poroshell 120 HILIC-Z column (2.1 × 150 mm, 2.7 µm, Agilent). An Agilent 6550 Series quadrupole time-of-flight mass spectrometer running in negative extended dynamic range mode was used to analyze the samples. Mobile phases were 10 mM ammonium acetate pH 9 in water with 5 µM medronic acid, and 85:15 acetonitrile:water with 10 mM ammonium acetate pH 9. A flow rate of 250 µL/min was used with a total measurement duration of 30 min. Mobile phase compositions were set as described in *Supplementary file 3*. Online mass-axis correction was performed using purine and Hexakis. Drying gas was provided at 13 L/min at a temperature of 22°C. Sheath gas was provided at 12 L/min at a temperature of 350°C. Capillary and nozzle voltages were set to 3500 and 0 V, respectively. Data analysis was performed in Agilent MassHunter Quantitative Analysis (version B.07.00) with peaks chosen based on retention time matching to compounds in standard solution. Twenty parts per million m/z windows were used for peak selection, with integration performed using the Agile2 algorithm. Average log2 transformed fold-changes were calculated between samples. Peak areas for the effect of rapamycin on metabolite levels and for the effect of nitrogen source downshifts are provided in *Supplementary files 4 and 5*.

## Yeast cultivation for growth rate and lag-time determination

Defined media with glucose and proline as carbon and nitrogen sources (see above) were inoculated with yeast to a target starting $OD_{600}$ of 0.05. Yeast were cultivated at a 1 mL volume within 48-well flower plates (M2P labs: MTP-48-B). The yeast were allowed to grow at 30°C for 18 hr at a shaking speed of 800 rpm with optical density monitoring every 5 min using a Biolector 1 microfermentation system. After 18 hr of growth, 1 mL of SD media with ammonium as a nitrogen source (see above) was added and the growth was tracked for another 24 hr. Data was smoothed by applying a 1 hr moving-window averaging to the recorded optical density. The data was natural log transformed, and the maximum slope was determined as well as the cultivation time required to reach the maximum slope. These values were recorded as growth rate and lag time. Average values from six technical replicates were taken per treatment and expressed as ratio to the average value for that experiment. These ratios are from four independent experiments in order to generate the data depicted in *Figure 3*.

## Protein extraction and peptide preparation for proteomics

The indicated yeast mutants were cultivated in a 96-well format in SD media (5 g/L ammonium sulfate [Sigma-Aldrich: A4418], 1.7 g/L Yeast Nitrogen base [BD Biosciences: 233530], 20 g/L D-(+)-glucose [Sigma-Aldrich: G8270]). 1 mL cultures were inoculated with the indicated mutant strains carrying the pHLUM prototrophy restoration plasmid and were allowed to double at least twice to achieve an

OD$_{600}$ of approximately 0.8. At this point, the yeast were treated with either rapamycin (400 ng/mL) or a vehicle control. After 1 hr of growth, the yeast were pelleted through centrifugation for 2 min at 2250 rcf in a 4°C precooled centrifuge. The supernatants were discarded and the remaining pellets were subjected to bead beating (425–600 μM diameter) for a duration of 20 min at 4°C after resuspension in 200 μL of protein extraction solution (8 M urea, 50 mM Tris [pH 8], 75 mM NaCl, 1 mM EDTA [pH 8]). After bead beating, extractions were supplemented with Triton X to 1% (w/v), dithiothreitol (DTT) to 5 mM, and sodium pyrophosphate to 10 mM. DTT-treated extracts were allowed to incubate at 30 min at 55°C. Samples were then alkylated with a final concentration of 10 mM iodoacetamide in the dark for 30 min. 50 μg of the extracted protein was then subjected to chloroform precipitation, and the clean protein extracts were subjected to tryptic digestions at a ratio of 50 μg of extracted protein to 1 μg of protease. Digestions were allowed to proceed for 16 hr at 37°C. Proteolysis was quenched through acidification with HCl.

## Quantitative proteomics

Peptides were analyzed by LC-MS/MS. On-line reversed phase chromatography was performed using a Vanquish Neo UPLC system (Thermo Scientific, Sunnyvale) equipped with a heated column compartment set to 50°C. Mobile Phase A consisted of 0.1% formic acid (FA) in water, while Mobile Phase B was 80% acetonitrile in water and 0.1% FA. Peptides (~1 μg) were loaded onto a C18 analytical column (500 mm, 75 μm inner diameter), packed in-house with 1.8 μm ReproSil-Pur C18 beads (Dr. Maisch, Ammerbuch, Germany) fritted with Kasil, keeping constant pressure of 600 bar or a maximum flow rate of 1 μL/min. After sample loading, the chromatographic gradient was run at 0.3 μL/min and consisted of a ramp from 0 to 43% Mobile Phase B in 70 min, followed by a wash at 100% Solution B in 9 min total, and a final re-equilibration step of three-column volumes (total run time 90 min).

Peptides from each sample were analyzed on an Orbitrap HF-X mass spectrometer (Thermo Fisher Scientific, San Jose, CA) using an overlapping window data-independent analysis (DIA) pattern described by *Searle et al., 2018*, consisting of a precursor scan followed by DIA windows. Briefly, precursor scans were recorded over a 390–1010 m/z window using a resolution setting of 120,000, an automatic gain control (AGC) target of 1e6, and a maximum injection time of 60 ms. The RF of the ion funnel was set at 40% of maximum. A total of 150 DIA windows were quadrupole selected with a 8 m/z isolation window from 400.43 m/z to 1000.7 m/z and fragmented by higher-energy collisional dissociation, HCD (NCE = 30, AGC target of 1e6, maximum injection time 60 ms), with data recorded in centroid mode. Data was collected using a resolution setting of 15,000, a loop count of 75, and a default precursor charge state of +3. Peptides were introduced into the mass spectrometer through a 10 μm tapered pulled tip emitter (Fossil Ion Tech) via a custom nano-electrospray ionization source, supplied with a spray voltage of 1.6 kV. The instrument transfer capillary temperature was held at 275°C.

All Thermo RAW files were converted into mzML format using the ProteoWizard package (version 3.0.2315; *Chambers et al., 2012*). Vendor-specific peak picking was selected as the first filter, and demultiplexing with a 10 ppm window was used for handling the overlapping window scheme. Processed mzML files were then searched using DIA-NN (version 1.8; *Demichev et al., 2020*) and the UniProt *Saccharomyces cerevisiae* proteome (UP000002311, June 15, 2021) as the FASTA file for a 'library-free' deep neural network-based search approach. Data was searched using deep learning-based spectra and retention time as described by *Demichev et al., 2020*, with trypsin as the protease, and allowing for X missed cleavages, with N-terminal methionine cleavage, and cysteine carbamidomethylation. Peptide length was allowed to range from 7 to 30 amino acids with a precursor charge state range from +1 to +4, a precursor range of 300–1800 m/z, and a fragment ion range of 200–1800 m/z. Data was processed to a 1% precursor-level false discovery rate (FDR) with mass accuracy, MS1 accuracy, and match between runs set to the software default settings. A single-pass mode neural network classifier was used with protein groups inferred from the input *Saccharomyces cerevisiae* FASTA file. Protein quantities were quantile normalized (*Bolstad, 2021*) and subjected to differential analysis as described above. The effect of normalization on data quality is visualized in *Figure 4—figure supplement 2*. GO term enrichment was performed using the clusterProfiler R package (*Yu et al., 2012*).

## Western blot for SCH9 phosphorylation

15 mL of wild-type and *cff1* yeast cells were grown to $OD_{600}$ of approximately 0.7 in defined media with glucose and ammonium as carbon and nitrogen sources. The cells were then exposed to either a vehicle control or 400 ng/mL rapamycin and were cultivated for a further 30 min. 9 mL of the yeast cultures were mixed with 1 mL of 100% (w/v) TCA. Cells were cooled on ice for 10 min prior to harvesting the cells by centrifugation (2 min at 1620 rcf). The resulting pellet was washed twice in 100% (v/v) acetone prior to resuspension in 100 µL of lysis buffer (50 mM Tris-HCl pH 7.5, 5 mM EDTA, 6 M urea, 1% [w/v] SDS). Cells were lysed by bead beating at 4°C for 20 min. Lysates were then incubated at 95°C for 5 min. Prior to gel loading, 200 µL of protein sample buffer containing 25% (v/v) β-mercaptoethanol was added to each sample before they were again incubated at 95°C for 5 min. Samples were subjected to a two-antibody Western blot analysis for SCH9-P/SCH9 quantification. Protein lysates were separated by SDS-PAGE on a 7.5% (w/v) gel. Membranes for all Western blot analyses were blocked and incubated with PBS-Tween 0.05% (v/v) and 5% (w/v) BSA. The antibodies used in this study were rabbit polyclonal anti-SCH9 (homemade, 1:10,000), mouse monoclonal anti-P-SCH9S-758 (homemade, 1:7000), and the corresponding fluorescent dye-coupled secondary antibodies (Alexa Fluor conjugated secondary antibodies, LI-COR). Protein was transferred to a nitrocellulose membrane, which was probed overnight with primary antibodies at 4°C. The membrane was then washed, and an incubation with the secondary antibodies was performed for 45 min at room temperature. Membrane development was performed using the Odyssey imaging system (LI-COR). Result quantification was performed using ImageJ (*Schneider et al., 2012*). Min-max normalization was performed for each sample through comparison to a standard sample generated from BY4741 yeast (MATa *his3Δ1, leu2Δ0, met15Δ0, ura3Δ0*) grown at 30°C in YPD and extracted according to the same protocol.

## Acknowledgements

We thank M Peter for the provision of yeast strains. Funding for PFD was provided by the ETH Fellows program with funding for DHS provided by the Human Frontiers Science Organization Long Term Fellowship program. Biorender.com was used to produce *Figure 1A*.

---

## Additional information

### Funding

| Funder | Grant reference number | Author |
| --- | --- | --- |
| ETH Zürich | Fellowship 19-FEL-12 | Peter F Doubleday |
| Human Frontier Science Program | Long Term Fellowship LT000604/2017-L | Duncan Holbrook-Smith |

The funders had no role in study design, data collection and interpretation, or the decision to submit the work for publication.

### Author contributions

Stella Reichling, Formal analysis, Investigation, Visualization, Writing - review and editing; Peter F Doubleday, Investigation, Visualization, Writing - review and editing; Tomas Germade, Formal analysis, Investigation, Writing - review and editing; Ariane Bergmann, Investigation, Writing - review and editing; Robbie Loewith, Supervision, Writing - review and editing; Uwe Sauer, Supervision, Writing - original draft, Writing - review and editing; Duncan Holbrook-Smith, Conceptualization, Formal analysis, Supervision, Investigation, Writing - original draft, Writing - review and editing

### Author ORCIDs

Peter F Doubleday  http://orcid.org/0000-0002-1784-1282
Tomas Germade  http://orcid.org/0000-0002-5144-1266
Robbie Loewith  http://orcid.org/0000-0002-2482-603X
Duncan Holbrook-Smith  http://orcid.org/0000-0003-0787-3165

Decision letter and Author response
Decision letter https://doi.org/10.7554/eLife.84295.sa1
Author response https://doi.org/10.7554/eLife.84295.sa2

## Additional files

### Supplementary files

- Supplementary file 1. Gene IDs for TOR and receptor-related mutant libraries.
- Supplementary file 2. Annotated ion intensities and sample information for flow-injection analysis.
- Supplementary file 3. Mobile phase composition for liquid chromatography-mass spectrometry (LC-MS) metabolomics.
- Supplementary file 4. Liquid chromatography-mass spectrometry (LC-MS) sample information and peak areas for rapamycin-treated wild-type yeast.
- Supplementary file 5. Liquid chromatography-mass spectrometry (LC-MS) sample information and peak areas for wild-type and *cff1* yeast during a nutrient downshift.
- MDAR checklist

### Data availability

Raw mass spectrometry are deposited at http://massive.ucsd.edu. For flow-injection raw data is found within dataset MSV000089939. LC-MS metabolomics raw data are deposited under ID MSV000089934. Raw and processed mass spectrometry data from the proteome analysis performed in Figure 3 are deposited under ID MSV000089935. Raw and processed mass spectrometry data from the proteome analysis performed in Figure 4 are deposited under ID MSV000089940. Processed metabolomics data are available within the supplemental materials or for visualization at https://rapamycin-yeast-metabolome.herokuapp.com/.

The following datasets were generated:

| Author(s) | Year | Dataset title | Dataset URL | Database and Identifier |
|---|---|---|---|---|
| Reichling S, Holbrook-Smith D | 2022 | High-throughput metabolome profiling of the response of 164 yeast mutants to rapamycin | https://massive.ucsd.edu/ProteoSAFe/dataset.jsp?task=e898821c968b4a85b56ee47fe7dcbc25 | MassIVE, MSV000089939 |
| Holbrook-Smith D | 2022 | LC-MS analysis of yeast metabolome extracts under rapamycin treatment and nitrogen source downshift | https://massive.ucsd.edu/ProteoSAFe/dataset.jsp?task=e26b83916d3840ba8f310955072da961 | MassIVE, MSV000089934 |
| Holbrook-Smith D, Doubleday P | 2022 | Proteomic analysis of yeast mutants treated with rapamycin | https://massive.ucsd.edu/ProteoSAFe/dataset.jsp?task=f9d182809f2d40c58a15e7ca6ce73541 | MassIVE, MSV000089935 |
| Holbrook-Smith D, Doubleday P | 2022 | Proteomic analysis of a CFF1 deletion mutant under rapamycin treatment | https://massive.ucsd.edu/ProteoSAFe/dataset.jsp?task=3475e614ee764825818f0935b51f51eb | MassIVE, MSV000089940 |

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
