## [Editor Report]

This work measures time-resolved metabolomes of 164 yeast mutants using a high-throughput method (FIA-MS). The dynamic, nontargeted measurements allow for an improved inference of gene function in the TOR pathway after a rapamycin treatment, including the annotation of three new genes in the TOR signaling pathway. This case study opens an avenue for combined studies of functional genetics and metabolism.

---

## [Decision Letter]

[Editors' note: this paper was reviewed by Review Commons.]

---

## [Author Response]

We sincerely thank the reviewers for their thoughtful and helpful input. Below were have included our response to the points raised by our reviewers.

Reviewer #1 (Evidence, reproducibility and clarity (Required)):Summary:Reichling et al. used flow-injection time-of-flight mass spectrometry (FIA-MS), a chromatography-free high-throughput method of untargeted metabolomics, to assess the temporary changes in the concentrations of polar metabolites within budding yeast exposed to the TORC1 (target of rapamycin complex 1) inhibitor rapamycin. The water-soluble metabolomes of the rapamycin-treated wild-type (WT) strain, many mutants in the non-essential genes lacking known protein components of the TORC1-upstream and -downstream signaling and numerous gene-deletion mutants missing the redundant small-molecule protein receptors that might participate in TORC1 signaling were analyzed by FIA-MS. The quality of the quantitative metabolome profiling performed by the FIA-MS method was validated with the help of a time-consuming liquid chromatography/mass spectrometry-based metabolomics of WT cells.Using the FIA-MS-based metabolome profiling, the authors revealed that rapamycin treatment upregulates or downregulates the polar metabolomes specific to a distinct set of cellular processes. Similar patterns of the temporal dynamics of rapamycin-induced changes in the metabolomes characteristic of these cellular processes were observed in WT cells and mutants deficient in known protein components of TORC1 signaling. Remarkably, the authors found that three mutants (i.e., BCK1, CLA4 and CFF1) impaired in the small-molecule protein receptors that were unknown for their roles in TORC1 signaling exhibit the temporal dynamics of rapamycin-induced metabolome changes comparable to the mutants deficient in the known positive protein regulators of TORC1 signaling. Furthermore, the comparative metabolome-based analyses of relationships between rapamycin-treated cells defective in the many known and several unknown protein regulators of TORC1 signaling allowed authors to conclude that such analyses objectively reflect the functional connectivity of these protein regulators. Moreover, the authors provided evidence that comparing the metabolome profiles in rapamycin-treated WT and TORC1 signaling mutant cells enables the identification of new metabolic reactions and pathways affected by and/or contributing to the TORC1-dependent nutrient signaling network.In this proof-of-principle study, the authors also employed the liquid chromatography-tandem mass spectrometry for a quantitative proteomic comparison of rapamycin-treated WT cells and mutants impaired in the known and previously unknown protein components of TORC1 signaling. They found that rapamycin induces comparable changes in the proteomes of all these cells. This proteomic analysis confirmed that the small-molecule protein receptors previously unknown as TORC1 signaling components and identified as such components only with the help of dynamic metabolome are integrated into the TORC1 signaling network. The authors further confirmed the essential contribution of the novel protein components of the TORC1 signaling to this type of nutrient signaling in the experiments on measuring growth rate changes following a nitrogen source upshift and assessing metabolome and proteome alterations after a nitrogen source downshift.Comments:The manuscript is clearly written and of high technical quality. All claims are convincing, fully supported by the experimental data and appropriately discussed in the context of previous literature. The authors have been fair in their treatment of previous literature. They provided the methodological detail sufficient for others to reproduce the experiments. No additional experiments are needed to support the claims made in the manuscript. The experiments are adequately replicated and statistical analysis is appropriately performed.Reviewer #1 (Significance (Required)):This pioneering study represents a major conceptual advancement in the fields of using condition-specific, dynamic metabolome profiling for the deep understanding of relationships between genes integrated into a signaling pathway, discovering novel genes incorporated into a cellular signaling network and its small-molecule regulators, and identifying the metabolic pathways governed by a signal transduction network.The treatment of the existing literature on the research topic by the manuscript's authors is balanced and fair.This well-organized and clearly written manuscript is a must-read for anyone interested in the molecular mechanisms of cellular signaling.My field of expertise involves exploring the molecular dynamics of complex cellular processes using advanced genetic, cell biological and biochemical approaches (including the mass spectrometry-based analyses of the cellular proteomes, lipidomes and water-soluble metabolomes).I recommend accepting this manuscript for publication in any journal affiliated with Review Commons.

We thank reviewer 1 heartily for their comments and for the time they spent reviewing our work.

Reviewer #2 (Evidence, reproducibility and clarity (Required)):In this manuscript, Rechiling et al., have used a unique approach by exploiting a temporal dynamic high-throughput metabolome profiling (using flow-injection time-of-flight mass spectrometry) to measure the metabolome profiles of many mutants in yeast, which allow them to newly annotate 3 genes in the TOR signaling pathway. This work demonstrate elegantly that dynamic perturbation of the cell allows inferring gene function when using a metabolomics-based guilt-by-association scheme. They were able to successfully find genes like CFF1, BCK1 and CLA4 which might act as positive regulators in the TOR pathway.This an interesting study since it provides an alternative approach to annotate gene function and their contribution to known signaling pathways by analyzing the dynamic of the soluble metabolome. Overall, the manuscript was concisely well written, and the finding has a great potential to improve our understanding of gene function and genetic determinism of metabolism in model organisms.Major comments:As the cellular response to rapamycin is not restricted to changes at the level of the metabolome, the authors investigated the proteomic response of each mutant to reaffirm their functional relationship to the TOR pathway. In this regard, it is not clear why the author did not consider a time-course analysis of the proteome as they did for the metabolome. The measurable steady-state proteomic signature might also reflect a buffered cellular state which might hide other response.

A time course analysis of the proteome for all mutants would be extremely interesting. However, due to the time and cost of proteomic analysis we instead focused on one time point for rapamycin treatment in the context of proteomics analysis. The proteome response data from mutants treated with rapamycin for 1 hour was used to identify the altered proteome responses of the mutants presented in figure 3A. This allowed us to explore the proteome responses of the mutants to rapamycin, and overcome the cellular buffering that our reviewer correctly points to.

Minor comments:Many typos in the methodology section (e.g., potassium phosphate.…)

The typos in the methods section have been corrected.

Supplementary Figure 1: Not clear what this figure show and how it supports the author's claim that FIA-MS was validated by LC-MS.

The positive linear relationship between the values for the measured metabolites by FIA-MS and LC-MS indicate that similar results were obtained with both methods. We have altered the text to make this clearer.

Reviewer #2 (Significance (Required)):This an interesting study since it provides an alternative approach to annotate gene function and their contribution to known signaling pathways by analyzing the dynamic of the soluble metabolome. The finding has a great potential to improve our understanding of gene function and genetic determinism of metabolic adaptation in model organisms.

We thank reviewer 2 for their helpful comments, and the time they spent reviewing our work.